# Expansion Control of Alkali-Activated Materials Using Waste Glass Cullet from Photovoltaic Panels as Fine Aggregates

**DOI:** 10.3390/ma17194902

**Published:** 2024-10-06

**Authors:** Ryo Yamanouchi, Kentaro Yasui, Hiroshi Yamada, Takayuki Fukunaga, Hideki Harada

**Affiliations:** 1Advanced Civil Engineering, National Institute of Technology, Kagoshima College, 1460-1 Shinko, Hayato-cho, Kirishima 899-5193, Japan; b18208@kagoshima.kosen-ac.jp; 2Department of Urban Environmental Design and Engineering, National Institute of Technology, Kagoshima College, 1460-1 Shinko, Hayato-cho, Kirishima 899-5193, Japan; 3Department of Civil and Environmental Engineering, Faculty of Engineering, Osaka Sangyo University, 3-1-1 Nakagakiuchi, Daito 574-8530, Japan; h-yamada@ce.osaka-sandai.ac.jp; 4Department of Civil Engineering, Faculty of Engineering, Kyushu University, 744 Motooka, Nishi-ku, Fukuoka 819-0395, Japan; fukunaga@doc.kyushu-u.ac.jp; 5Solar Lab, Solar Frontier K.K., 1815 Tajiri, Kunitomi-cho, Miyazaki 880-1104, Japan; hideki.harada.4750@solar-frontier.com

**Keywords:** alkali-activated material, sodium orthosilicate, photovoltaic panel glass cullet, alkali–silica reaction

## Abstract

Glass cullet (GC) generated from the disposal of photovoltaic (PV) panels are typically landfilled, and effective GC utilization methods must be established for PV generation. In this study, alkali-activated material (AAM) mortars were prepared from the paste of fine blast-furnace slag powder, fly ash, and sodium orthosilicate (SO) and mixed with crushed sand and GC to investigate the potential use of GC as a fine aggregate in AAM. The replacement of crushed sand with GC did not considerably affect the flowability of the mortar, whereas the compressive strength decreased with the increasing GC replacement rates. Although expansion due to the alkali–silica reaction (ASR) was observed in mortars wherein GC replaced crushed sand, the expansion can be controlled by reducing the amount of mixed SO, autoclaving the GC, performing preleaching to remove the Si that causes the ASR, and replacing the blast-furnace slag with fly ash. By enforcing measures against the expansion, the possibility of using GC as fine aggregate is enhanced considerably, thus increasing the feasibility of continuous PV production.

## 1. Introduction

Photovoltaic (PV) power generation is the third most extensively used renewable energy source after hydropower and wind power, with the number of worldwide installations increasing considerably in recent years [1]. However, the emissions of the PV panels (PVPs) that have reached the end of their useful life (of approximately 25 years after installation) are expected to increase rapidly from 2030, with a cumulative PVP-disposal volume of 60–78 billion kg by 2050 [2,3,4]. PVP recycling is essential from the perspective of nature conservation [3] and aims to establish methods for the effective use of glass, which constitutes the majority of PVPs. However, PVPs contain many hazardous substances, such as Pb and Cd, which limit their reusability [5]. Therefore, only approximately 10% of the world’s PVP waste is currently reused [6], with waste glass being disposed of in landfills after crushing [7,8]. The current disposing methods of PVP waste may fill up space at the future final disposal sites, thus presenting various concerns. Furthermore, toxic substances seeping into the ground may affect the environment and human health [9]. Based on the current state of PVP disposal, the European Union adopted PV-specific waste regulations for the first time in 2012, requiring PVP manufacturers to collect and dispose of PVPs after usage [4,10]. However, most countries, including China, the United States of America, and Japan, do not have specific legislation for the disposal of PVPs [10]. Additionally, the expected required time for the processing of all PV waste using current technology and the processing capacity is in the order of decades [2,11]. Therefore, establishing a glass cullet (GC) recycling system is an immediate requirement for the continued usage of PV systems.

Research has been conducted on the use of GC as a concrete fine aggregate in the construction sector. However, because glass contains abundant silica components, an alkali–silica reaction (ASR) occurs [12]. Conversely, ASR prevention techniques, such as the carbonation curing of glass aggregate-mixed mortar, have been reported to densify the surface, reduce the pH of the pore solution [13], and passivate the aggregate surface by adding calcium nitrate [14]. Moreover, several studies have focused on the use of glass fines as a substitute for cement [15] and aluminosilicate precursors in alkali-activated material (AAM) [16]. These studies employed glass as a substitute and a precursor, albeit only at low substitution rates and in small quantities. Therefore, identifying methods that can promote mass GC utilization is crucial.

AAMs have gained considerable attention worldwide as alternative concrete materials to cement. Although concrete is the most extensively used construction material [17,18,19], its primary constituent, cement, emits large amounts of CO_2_ during production, accounting for 5–7% of the world’s CO_2_ emissions [20,21,22]. Therefore, major changes must be implemented in the construction industry to meet the target set in the Paris Agreement, that is, to limit the temperature increase from pre-industrial times to within 1.5 °C [23]. Davidovits proposed the use of geopolymer as an alternative cement material [24] in France in the 1970s. In the 1980s, a solidified product was formed following the condensation polymerization of aluminosilicate precursors with highly alkaline solutions, such as NaOH/KOH and water glass [25,26]. Furthermore, recent studies have shown that the major binding phase of geopolymer is sodium (calcium) aluminosilicate hydrate (N-(Ca-)ASH) as a low-Ca AAM [27,28]. AAM does not use cement and can utilize industrial byproducts as precursors, which reduce the CO_2_ emissions during production by approximately 80% when compared with cement-based concrete [29]. Therefore, using AAM concrete as a construction material can significantly reduce CO_2_ emissions in the construction sector. AAM can be classified by name according to the mixing ratio of precursors [30,31,32]. Accordingly, the precursors formed with fly ash (FA) and alkaline solution are termed alkali-activated fly ash (AAFA). Those formed with FA, blast-furnace slag fine powder (BFS), and alkaline solutions are termed alkali-activated fly ash/slag (AAFS), whereas those formed with BFS and alkaline solution are termed alkali-activated slag (AAS).

In addition to the significant reductions in CO_2_ emissions, AAM presents high heat resistance. Typically, cement concrete involves dehydration and decomposition of cement hydrates at temperatures above 500 °C, causing changes in the pore structure and a significant reduction in strength. However, AAM produces small amounts of cement hydrates; therefore, dehydration and decomposition are unlikely to occur, even in high-temperature environments, and AAM can be used as a heat-resistant material [33]. Along with its heat resistance, AAM has an extremely high acid resistance when compared with cement [34]. This is because cement hydrates are composed of Ca(OH)_2_ and C–S–H. Thus, the reaction product is dissolved using acid, whereas in AAM, the product is an aluminosilicate condensation polymer. Consequently, the dissolution of the acid is unlikely to occur. Even in the AAFS case, the acid resistance is not significantly reduced up to a substitution rate of approximately 20% for FA [35]. Other studies have indicated that heavy metals—such as Cd and Pb—and radioactive substances—such as Sr and Cs—can be fixed in AAM [36,37]. Additionally, the Ca content of AAM is lower than that of cement [38]. Most of the alkali is used for initial strength development, and the reactive Si in reactive aggregates is used for strength development, resulting in diminished expansion owing to ASR development compared with the case in which cement is used [39,40,41,42]. Thus, the use of aggregates that exhibit reactivity to ASR is expected.

In recent years, the shortage of sand resources required for infrastructure development has become a significant problem worldwide [43], and the creation of alternative sand sources is an immediate requirement [44]. Therefore, PVP-derived GC can be used as an alternative to AAM, which is less prone to ASR, to alleviate the shortage of sand resources and the continuous usage of PV power generation. This can also contribute to the development of an environmentally friendly recycling industry, as well as a low-carbon and resource-recycling society.

In this study, the objective is to establish an effective method for the utilization of waste glass from PVPs and the creation of alternative aggregates. First, an AAM mortar was prepared by replacing crushed sand with the GC derived from PVPs (Figure 1) as a fine aggregate. Although NaOH is often used as an alkaline source for conventional AAM, sodium orthosilicate (SO) (Figure 2) has been used as an alternative to NaOH for safety reasons during production. The flow value, compressive strength, and expansibility of the AAM mortars prepared with BFS or FA as precursors were investigated, and the effects of the replacement ratio of GC on these physical properties were experimentally clarified. Furthermore, based on the results of the expansibility of the AAM mortars, the ASR gel formation status of some specimens was visualized by using polarizing and scanning electron microscopy. From these results, the usability of GC (as a fine AAM aggregate) and the method for suppressing ASR expansion (when used as an aggregate) are reported.

## 2. Experimental Overview

### 2.1. Materials and Specimen Preparation

Table 1 lists the specifications of the raw materials used in this study. SO was used as the alkali source, and BFS and FA were used as precursors. Crushed sand, which was reported to be nonreactive to ASR, was used as the fine aggregate, and a prescribed amount of GC was substituted for the crushed sand. The GC was made from crushed glass that was removed during the dismantling of the solar PVPs.

Table 2 lists the chemical compositions of the raw materials. An X-ray fluorescence analyzer (ZSX Primus II, Rigaku Corporation, Akishima, Japan) was used for the chemical composition analyses. The primary components of the dried SO are Na and Si, which contain 1.7% water. The hydrolysis of SO produces sodium metasilicate and NaOH. Therefore, the reaction mechanism is similar to that of conventional AAM using NaOH and waterglass, owing to the reaction between water and SO during kneading. The BFS contained more Ca, Si, and Al, whereas the FA contained more Si and Al. FA is used in AAM as a precursor along with BFS; however, FA has a lower Ca content than BFS. The GC contained more Na and Ca, along with Si and 0.22% Sb.

Table 3 lists the mix designs for the AAM mortar formulations used in this study. In Series 1, the volume ratio of fine aggregate to AAM paste was 1.0, that of the alkaline solution to the precursor ((W + SO)/P) was 1.5, and the molar ratio of Na to water in SO (alkali–water ratio, Na/H_2_O) was 0.09. Further details are presented in Section 3, which follows the results obtained in Series 1. New experiments were conducted using the formulations of Series 2–4. For these series, the volume ratios of fine aggregates to AAM paste were set to 1.0, (W + SO)/P to 1.4, and Na/H_2_O to 0.065. In Series 3, the GC was autoclaved to preleach Si from the GC before mixing.

NaOH is typically used as an alkali source for AAM [24,25,26,34]. However, in this study, SO was used as an alternative alkali source to the deleterious NaOH. Nevertheless, the value of (W + SO)/P was 1.4 or 1.5 (the mass ratio was in the range of 0.52–0.58), which is similar to the ratios proposed in previous studies [45,46,47].

BFS and FA were used as precursors to prepare two types of AAM mortar: 100% BFS (AAS) and a 50% mixture of BFS in Series 1–3 and FA (AAFS) in Series 4.

For the fine aggregates, the percentages of GC in the aggregate were 0%, 25%, 50%, 75%, and 100%. The blends are referred to as 1-G0 to 1-G100, 2-G0 to 2-G100, 3-G0 to 3-G100, and 4-G0 to 4-G100 based on the series and GC replacement percentages.

To prepare the specimens, the precursors, fine aggregates, and SO were first added to a mortar mixer with a capacity of 5 L and mixed at 11.5 rpm for 2 min. All the water was then added, mixed at 65.0 rpm for 1 min, and allowed to stand for 30 s to scrape off any mortar remaining on the mixer paddle. The materials were mixed during a secondary mixing cycle at 65.0 rpm for 1 min until homogeneity was achieved. Subsequently, the resulting material was cast into a cylindrical mold with an inner diameter of 50 mm and a height of 100 mm, vibration-compacted using a table vibrator for 1 min, sealed, and cured to the prescribed material age. At least three compressive strength specimens were used for each condition, and the test results were averaged over these specimens.

### 2.2. Experimental Methods

#### 2.2.1. Mortar Flow Test

To measure the mortar flowability, a flow test was conducted based on Japanese Industrial Standards (JIS) R 5201, where the mortar was packed into a flow cone (upper inner diameter: 70 mm, lower inner diameter: 100 mm, and height: 60 mm) immediately after mixing. After removing the cylinder, the length of the mortar that spread with 15 falling movements was measured in two directions, and the average was calculated. This procedure was performed thrice, and the average of these values was used as the flow.

#### 2.2.2. Compressive Strength Test

Compressive strength tests of the mortar were conducted according to JIS A 1108 (ISO 1920-4) at 1 d (demolding age), 7 d (shipping age: Series 1), and 14 d (shipping age: Series 2–4). Three specimens were subjected to compression tests for each mix design and material age, and the average values were calculated. The loading rate during the compressive tests was set at 36 MPa/min. The AAM produced in this study was intended for use in precast products, such as road curbs; therefore, the target compressive strength was set at 24 MPa.

#### 2.2.3. Accelerated Expansion Test

To analyze the effects of the GC replacement rate on the expansibility of the AAM mortars, accelerated expansion tests were performed in accordance with the American Society for Testing Materials (ASTM) standard C1260 method for each mix design, as shown in Table 3. Specifically, cylindrical specimens (50 mm [diameter] × 100 mm [height]) were sealed and cured at a constant temperature of 20 °C for 7 d for Series 1 and 14 d for Series 2–4, immersed in water at 80 °C for 24 h to confirm thermal expansion, and then immersed in a NaOH solution (1 mol/L) at 80 °C for 2–4 weeks for ASR-accelerated curing. The NaOH solution was changed after 3 and 7 d of immersion to ensure the consistency of the concentration, and 1 L per specimen was used for 2 weeks. The length measurements were performed using molded strain gauges (PMFL-50, Tokyo Measuring Instruments Laboratory Co., Ltd., Shinagawa-ku, Japan). The length changes in the strain gauges embedded in the specimens were measured every 30 min using a data logger (TDS-540, Tokyo Measuring Instruments Laboratory Co., Ltd., Shinagawa-ku, Japan). The average value of the two specimens for each mix design was used as the expansion rate for the mix designs.

#### 2.2.4. Elution of Si and Na Components from GC via Autoclaving and Analysis of Eluted Water

GC is a three-element material comprising Si, Na, and Ca and is considered to be responsible for the expansion of the AAM. An autoclave (20 L capacity, Nitto Kouatsu Co., Ltd., Tsukuba, Japan) was used to determine whether these elements could be recovered. In the tests, 11 kg of GC and 4 kg of distilled water were placed in the vessel as the soaking solutions, and the temperature was increased to a maximum of 180 °C under a nitrogen gas atmosphere and maintained at the maximum temperature for 3 h. The rate of temperature increase was 50 °C/h, and the internal pressure was 0.8 MPa.

To confirm the effect of autoclaving on component leaching from GC, the Si and Na concentrations were measured using inductively coupled plasma optical emission spectroscopy (ICP-OES; iCAP 6300 Duo, Thermo Fisher Scientific K.K., Minato-ku, Japan), whereas low Ca concentrations were measured using ICP mass spectrometry (ICP-MS; Agilent 7700X, Agilent Technologies Japan, Ltd., Hachioji, Japan).

#### 2.2.5. Observations on ASR Gel

Polarized light microscopy (ECLIPSE LV100N POL, Nikon Corporation, Shinagawa-ku, Japan) was performed to observe the micromorphology of the AAS matrix following the accelerated expansion test (14 d). The specimens were cut into separate parts with dimensions of 30 mm (length) × 20 mm (width) × 5 mm (height) using a precision cutting machine. Subsequently, they were polished using water-resistant abrasive paper in a dry process. These polished specimens were then further polished with fluid paraffin and attached to a microscope slide. Thus, the specimens were cut to a thickness of approximately 1 mm using a precision cutting machine, dry polished to a thickness of approximately 100 µm using water-resistant abrasive paper, and polished to a thickness of approximately 20 µm using fluid paraffin. In addition, thin mirror-polished specimens were prepared by dry polishing with diamond polishing powder.

Furthermore, the thin mirror-polished specimens of the ASR gel were imaged using scanning electron microscopy (JSM-IT-300HR, JEOL Ltd., Akishima, Japan) at an acceleration voltage of 15 kV. Additionally, the elemental compositions of the ASR gel were analyzed via energy-dispersive spectrometry (EDS). The specimens were subjected to carbon deposition to prevent antistatic properties.

## 3. Results and Discussion

### 3.1. Mortar Flow Value

Figure 3 depicts the relationship between the GC substitution rate and the flow value. The flow values slightly increased with the GC substitution when compared with those before the substitution; however, the difference was marginal. This can be attributed to the smooth surfaces and low water absorption of the GC compared with sand, which has a reduced effect on the flowability after displacement. Similar findings have also been confirmed in other studies [48,49].

The formulation with 50% BFS in precursors (Series 4; AAFS) exhibited superior flowability when compared with that with 100% BFS in the precursors (Series 1, 2, and 3; AAS). This can be attributed to the replacement of the BFS precursor with FA, which is a spherical particle that improves flowability. In Series 4, the flow value increased in the fine aggregate for GC ratios up to 50%, although it decreased thereafter. As the flow value is affected considerably by the size and density of the fine aggregate, the fluidity is believed to increase for GC ratio values up to 50%, owing to its replacement with GC (with a density of 2.46 g/cm^3^, which is lower than the sand’s density of 2.85 g/cm^3^). Moreover, as the replacement amount increased, the fluidity decreased owing to the GC (which comprises larger particle sizes than sand) occupying the mortar, thus making it difficult for the GC to move. The reason that this phenomenon did not occur in Series 1, 2, and 3 was presumably because the density of BFS (equal to 2.91 g/cm^3^)—which is the base of the mortar—is greater than that of FA (equal to 2.36 g/cm^3^); therefore, it is less affected by the aggregate compared with FA. This phenomenon is planned to be clarified in future experiments.

### 3.2. Mortar Compressive Strength

Figure 4 depicts the relationship between the compressive strength and GC replacement ratio for all tested series. The plotted values are the means, and the error bars denote single standard deviations above and below the means. Regardless of the mix design, the compressive strength decreased as the replacement ratio of GC to crushed sand increased. This is attributed to the smooth GC surfaces (Figure 1) that led to diminished adhesion between the paste and aggregate compared with the crushed sand case. In a previous study, waste glass from crushed food bottles was mixed with AAM as a fine aggregate [50]. The compression strength was also reduced when compared with that observed when sand was used as the fine aggregate. Furthermore, the increased smoothness of the waste glass could cause cracking and insufficient adhesion between the recycled glass waste and FA paste interface. This reduced compressive strength has also been reported in cases where AAM was mixed with waste cathode ray tube glass [49]. However, focusing on the age of the shipping material, the target compressive strength of 24 MPa was met in all the conditions associated with Series 1, 2, and 3. The GC had a fineness modulus of 4.73; however, a smaller particle size would reduce the smooth surface aggregates and can alleviate the reduction in compressive strength owing to the replacement of GC.

In Series 4, the target strength was achieved up to a GC replacement rate of 75%. For a material age of 1 d, compressive strengths greater than 5 MPa were obtained for all formulations, which is considered to allow demolding. Herein, focusing on the precursors, the strength of the BFS-only mortar (BFS/P = 100%, Series 2 and 3) exceeded that of the mortar mixed with BFS and FA (BFS/P = 50%, Series 4). The hardening of the AAM [25,51,52] was caused by the hydration reactions in the case of BFS-only mortar and by the combination of condensation polymerization and hydration reactions for a mixture of BFS and FA. The higher strength of the mortars with a higher proportion of BFS was attributed to the enhanced formation of Ca(OH)_2_ in the BFS mixture.

### 3.3. Change in the Expansion Rate

Figure 5 depicts the change in the expansion rate in Series 1 of the AAS mortars with different GC replacement rates over time. The measurement started with the specimen immediately after demolding (1 d of age) as the reference length, and shrinkage due to the hydration reaction of the AAM was observed up to 6 d after demolding (7 d of age). Subsequently, thermal expansion was observed owing to immersion in water at 80 °C for 1 d. Expansion due to immersion in the NaOH solution was observed 7 d after demolding. The NaOH solution was changed at 10 and 14 d after demolding, and expansion due to the change was observed because the decreased concentration of NaOH before the exchange returned to its original concentration after the replacement of NaOH; however, the change in the expansion rate decreased with age. Although the ASTM C1260 standard defines the expansion rates of ≥0.2% as being “potentially harmful,” AAM mortars with GC replacement rates in the range of 50–100% exhibited expansion rates of ≥0.2% within 1 week after immersion in a NaOH solution, and mortars with GC replacement rates of 25% exhibited expansion rates of ≥0.2% at 8 d after immersion in a NaOH solution. Therefore, ASR may have occurred in the AAM, as well as when cement was used. Conversely, the rate of expansion also increased with an increase in the GC replacement rate, indicating that the pessimum mixing rate was not present in the formulation of Series 1. 

In Series 1, where the target was planned to ship the specimens within 1 week, the AAM specimens were considered to contain an excessive amount of Na, causing expansion. This excess Na accelerated the expansion after immersion in the NaOH solution during the accelerated expansion test. Furthermore, the GC was rich in Na (Table 2); this increased the alkalinity of the interface between the GCs and AAM paste and promoted more expansion. Therefore, the amount of excess Na in the mortar must be reduced to control the expansion rate, which is desirable to reduce the amount of SO used. However, because AAM mortars have been reported to increase in strength at increasing alkali contents [30,53,54], reducing the amount of SO causes a reduction in strength. Therefore, a longer shipping age—to meet the target strength and the maximization of alkali consumption in the mortar for curing—would contribute to the control of post-service expansion. Therefore, mortars with controlled amounts of alkali were prepared in Series 2.

Figure 6 depicts the temporal changes in the expansion rates of the AAS mortars at different GC replacement rates in Series 2. Similar to Series 1, shrinkage due to the hydration reaction was observed up to 13 d after demolding (14 d of age), followed by thermal expansion due to immersion in water at 80 °C. Subsequently, expansion was observed owing to immersion in the NaOH solution, although the change was smaller than that in Series 1. At a GC substitution of 100%, several days elapsed after NaOH immersion to allow the expansion ratio to reach values ≥0.2% in Series 1; by contrast, in Series 2, it reached values ≥0.2% at 13 d after the onset of immersion, demonstrating that the reduction in the alkali content effectively controlled the expansion. However, a 100% GC replacement rate is still a “potentially detrimental” expansion rate with values >0.2%. Additional expansion control may be required for the full use of GC as a fine aggregate. The pessimum mixing ratio could also be absent in the Series 2 formulation because the expansion ratio increased as a function of the GC substitution ratio, as observed in Series 1. 

Thereafter, mortars based on the formulations of Series 3 and 4 were prepared according to the results. In addition to the reduction in the alkali content in Series 2, Series 3 was formulated using GC with preleached Si components, whereas Series 4 was formulated using FA mixed with precursors to reduce the amount of Ca contributing to expansion.

Figure 7 depicts the temporal variations in the expansion rates of the AAS mortars at different GC replacement rates in Series 3. In Series 2, the mortars with a 100% GC replacement ratio exhibited an expansion rate of >0.2%, whereas in Series 3, the expansion rate was <0.1% in all the formulations.

The changes in the expansion rate of Series 2 and 3 were compared with those of the AAS mortar prepared in a previous study [47]. In the previous study, ASR reactive aggregate was used as the fine aggregate, and the volume ratio of fine aggregate to AAM paste was 0.96, that of (W + sodium silicate)/P was 1.5, and that of Na/H_2_O was 0.06. Accelerated expansion tests were also performed by immersion in a NaOH solution (1 mol/L) at 80 °C; thus, the experimental conditions were considered to be approximately the same as those for Series 2 and 3. The AAS mortar of Series 2, shown in Figure 6, continued to expand for 21 d (35 d after demolding) from the start of expansion under all conditions. The expansion rate of 2-G100 reached 0.2% of Series 2, and the measurement ended at this point, suggesting that the expansion would continue. Furthermore, the AAS mortar of Series 3, shown in Figure 7, continued to expand for 2 d (16 d after demolding) from the start of expansion under all conditions. After reaching a maximum value, the expansion rate remained approximately constant for the next 19 d. In the previous study, the expansion rate continued for 28 d, reached 0.3%, and then remained approximately constant for the next 196 d. The expansion rate differed depending on the influence of the fine aggregate; however, once the expansion stopped, it tended not to expand again.

Figure 8 depicts the temporal variations in the expansion rates of the AAFS mortars at different GC replacement rates in Series 4. In Series 1 to 3, the pessimum mixing ratio was absent; however, in Series 4, the expansion rate was highest at a GC replacement ratio of 50%, indicating the presence of a pessimum mixing ratio. This can be attributed to the fact that the 50% GC replacement rate had a larger proportion of nonreactive aggregates and a smaller proportion of reactive aggregates in the fine aggregate when compared with 75% and 100% replacement rates, resulting in a higher amount of alkali acting on the individual particles of the reactive aggregate (i.e., GC). However, no formulation exhibited an expansion rate greater than 0.2%. In addition, under almost the same conditions as in this study, an accelerated expansion test was conducted on AAFA mortar, which uses 100% FA. Although there was a slight expansion from the start of the expansion, the expansion rate was very low at 0.025% after 224 d [47]. This can be attributed to the lower Ca content of FA compared with that of BFS (see Table 2), and the ASR does not occur when the Ca content in the precursor is low [55,56]. Therefore, the use of FA as a precursor can effectively reduce the expansion. 

### 3.4. Components of Soaking Solution

Table 4 presents the results of the ICP analysis of the components of the soaking solution that remained after autoclaving the GC. The GC was used in Series 3 after autoclaving. The initial Si, Na, and Ca concentrations in the distilled water (used as the GC soaking solutions) were 0, 0, and 0 mg/L, respectively. Conversely, the Si, Na, and Ca concentrations in the water after autoclaving were 1300, 1600, and 4.8 mg/L, respectively. The mass balance was calculated based on the GC used for autoclaving and the amount of distilled water; this means that 373 mg of Si, 458 mg of Na, and 1.38 mg of Ca were eluted per 1 kg of GC. The reason for the significant suppression of expansion in Series 3 is attributed to the prior recovery of Si, Na, and Ca, which contributed to the ASR from the GC. In particular, the amounts of Si and Na dissolved were remarkable, and the prior elution of Si and Na components from the autoclaved GC effectively controlled the expansion.

### 3.5. Images of the Cut Surface of the Specimen

Figure 9 presents the polarized light microscopy images of the cut surface of the specimen following ASR-accelerated curing of the 1-G100AAM mortar (14 d). Gelation progressed along the contours of the GC, and ASR expansion cracks were densely generated in the AAS paste from the gelatinized areas.

Figure 10 presents the polarized light microscopy images of the cut surface of the specimen following ASR-accelerated curing of the 3-G100AAM mortar (14 d). Only a few GC particles were gelatinized along the contours. However, certain GC particles exhibited gel veins generated inside the particles, which slightly extended into the paste as microcracks.

Figure 11 presents scanning electron microscopy images of the cut surfaces of the specimens and the elemental compositions of the ASR gels. In the 1-G100 specimen, gelation progressed along the contours of all GC particles, which partially crystallized in a rosette shape (Figure 11a). ASR expansion cracks were generated in the AAS paste from the gelatinized areas. The Na/Si atomic ratio calculated from the wt% values of SiO_2_, CaO, and Na_2_O contained in the S1-A gel obtained from EDS was 0.41, and the Ca/Si atomic ratio was 0.11. 

In the 3-G100 specimen, gelation progressed along the contours of some GC particles, and microcracks were observed from the gelatinized areas to the AAS paste (Figure 11b). The Na/Si and Ca/Si atomic ratios in the S3-A gel were 0.42 and 0.22, respectively. The Na/Si and Ca/Si atomic ratios of the ASR gel reported in a previous study [55] were in the ranges of 0.09–0.49 and 0.11–1.01, respectively; the values obtained in this study are within these ranges.

Figure 12 presents a scanning electron microscopy image of the interface between the GC and the paste on the cut surface of the 3-G100 specimen. Herein, the thin cracks in the images were generated when the scanning electron microscopy observation chamber was placed in a vacuum. These are not cracks caused by ASR.

The interface shown in Figure 12a was identified by determining the color difference using polarized light microscopy and backscattered electron imaging. The interface is located on the extension of the light blue arrow shown on the left and right parts of Figure 12b, and the composition of point S3-G1, located inside the edge of the GC, was similar to that of the GC, as summarized in Table 2. However, points S3-G2 and S3-G3, which are located at the edges of the GC, contained less Si and Na, although much more Ca than S3-G1. This difference was observed only in the 3-G100 specimen that had been autoclaved, and the area (with a thickness of approximately 2 μm) along the contours of all GC particles is believed to correspond to the area where Si and Na leached out. The Na content at points S3-P1 in the paste where ASR did not occur was lower than those at S1-A and 3-A where ASR occurred. The total oxide content at point S3-G1 was 94.3% because it was unhydrated, whereas the total oxide content at the other points was in the range of 60–70%, owing to the formation of hydrates.

This study demonstrated that pre-eluting the components contributing to the ASR from GC via autoclaving was the most effective method of controlling the AAM expansions. In practice, secondary concrete product manufacturers may own autoclaves; therefore, pretreating GC is considered possible for concrete manufacturers by using their autoclaves. However, the effective use of the eluate remains a problem; therefore, a method for reusing this eluate must be considered in future studies.

Conversely, when the amount of alkali is increased, and the pH of the pore solution becomes extremely high, ASR is thermodynamically inhibited. Additionally, Si dissolved from the aggregate surface may contribute to the hydration of the raw materials [55]. In this study, the amount of alkali was adjusted under two conditions using SO, although different results would have been obtained if the alkali concentration was higher.

In addition, this research is intended for use for precast products, such as road curbs. Therefore, since precast products are exposed to rainwater and soil water interference, which may leach heavy metals, it is necessary to conduct leaching tests for heavy metals.

## 4. Conclusions

In this study, AAM mortars were prepared using crushed GC as a fine aggregate and SO as an alkali source for the effective utilization of waste glass from landfilled PVPs, and the effects of the GC replacement ratio on their physical properties were investigated. The findings are summarized as follows:1.In this study, AAM was prepared using SO (as an alkali source), BSF or FA (as precursors), and crushed sand (as a fine aggregate). Accordingly, AAM with sufficient strength was obtained. In addition, the strength increased as a function of the SO content in AAM. Therefore, SO was confirmed to be a safe alkali source that can be used in AAM as an alternative to NaOH or water glass.2.The flow of the AAM mortar prepared in this study was in the range of 180–220 mm under all tested conditions, indicating good fluidity. In particular, the AAS using 100% BFS exhibited a minor increase in flow at increasing GC replacement ratios. This is because the surface of GC was smoother and less water absorbent than sand; therefore, the replacement had less prominent effects on fluidity.3.The AAFS using 50% FA yielded increased flow values owing to the replacement with GC, which was lighter than sand for GC ratio values up to 50% in the fine aggregate. However, the flow values decreased thereafter owing to the GC, which contained larger particle sizes than sand (which occupied the mortar) and made it difficult for the GC to move.4.The compressive strength of AAM mortars with partial replacement of fine aggregates with GC decreased as the replacement ratio increased. This is attributable to the smooth GC surfaces. However, a target strength of 24 MPa was achieved when the precursor was entirely composed of BFS. When the precursor composition contained 50% FA, the target strength reached a 75% GC replacement ratio.5.ASR expansion occurred in the AAS and AAFS mortars in which some of the fine aggregates were replaced by GC. The amount of expansion of the AAS mortar increased as the GC replacement rate increased and was different from that of nonreactive ASR aggregates. Conversely, the amount of expansion of the AAFS mortar was maximized at a GC replacement rate of 50%; thus, the presence of a pessimum mixing ratio was observed.6.Observation of the cut surface of the AAS mortar after the expansion test showed that the gelation progressed along the contours of the GC, and ASR expansion cracks were densely generated in the AAS paste from the gelatinized areas. Conversely, only a few GC particles were gelatinized along the contours on the cut surface of the AAS mortar mixed with the autoclaved GC.7.From the results presented above, the use of effective methods for suppressing ASR expansion (when GC is used as an AAM fine aggregate) is intended to reduce the amount of the alkali source when preparing AAS mortar, autoclave GC to leach out the components that contribute to ASR and use BFS and FA as precursors of AAM, with the autoclaving GC method being the most effective.

Using GC from PVPs as fine aggregates for AAM by considering the suppression of expansion can increase the feasibility of continued PV production. The effective usage of the autoclave eluate remains a problem; therefore, a method for reusing this eluate must be considered in future studies. 

## Figures and Tables

**Figure 1 materials-17-04902-f001:**
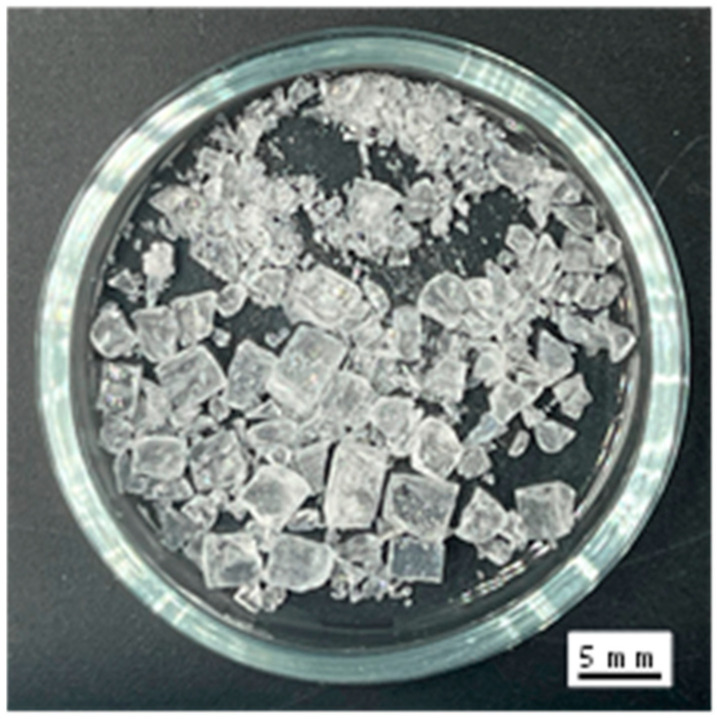
Glass cullet.

**Figure 2 materials-17-04902-f002:**
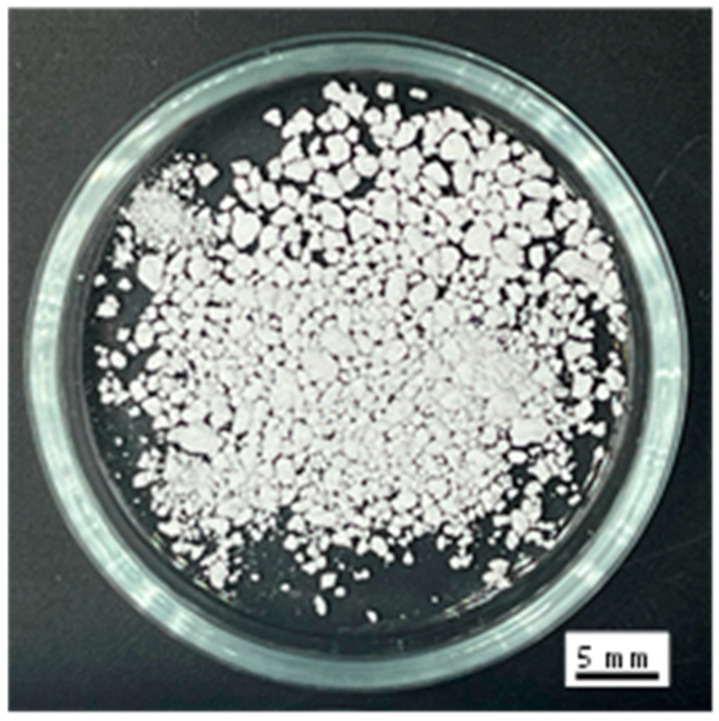
Sodium orthosilicate.

**Figure 3 materials-17-04902-f003:**
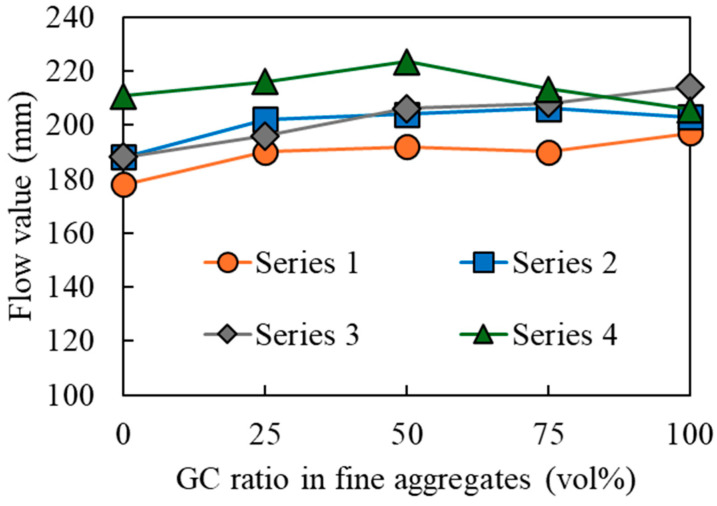
Relationship between the glass cullet (GC) substitution rate and flow value.

**Figure 4 materials-17-04902-f004:**
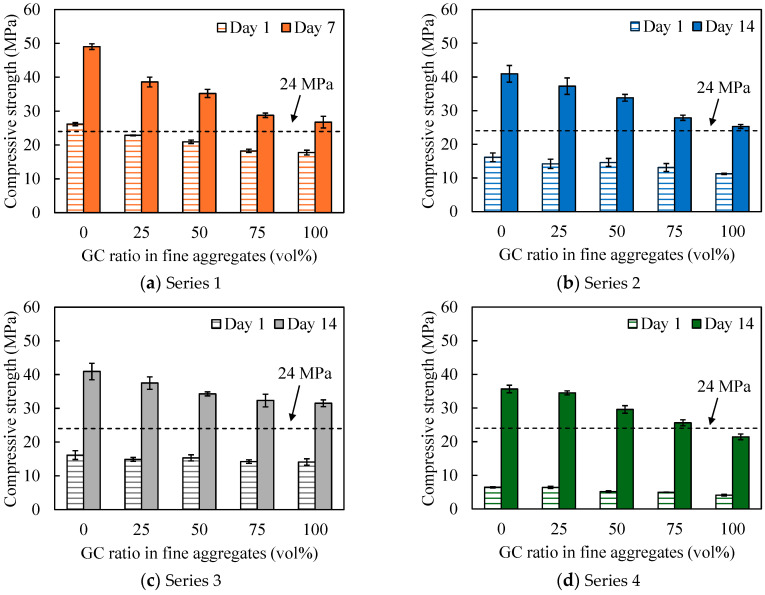
Relationship between the GC replacement rate and compressive strength.

**Figure 5 materials-17-04902-f005:**
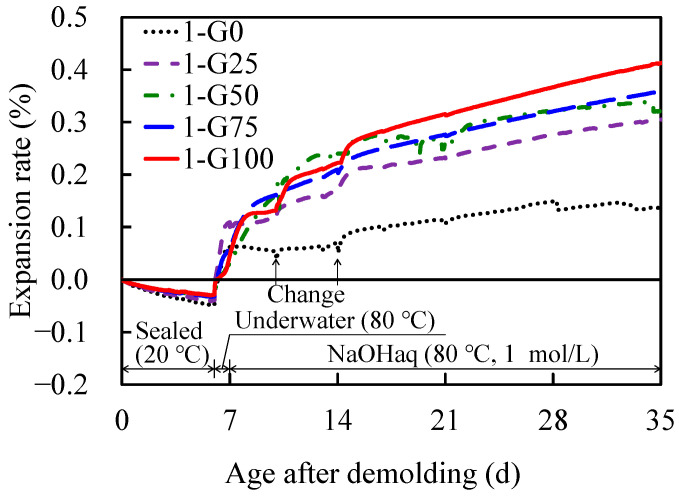
Diurnal changes in expansion rate as a function of demolding age (Series 1).

**Figure 6 materials-17-04902-f006:**
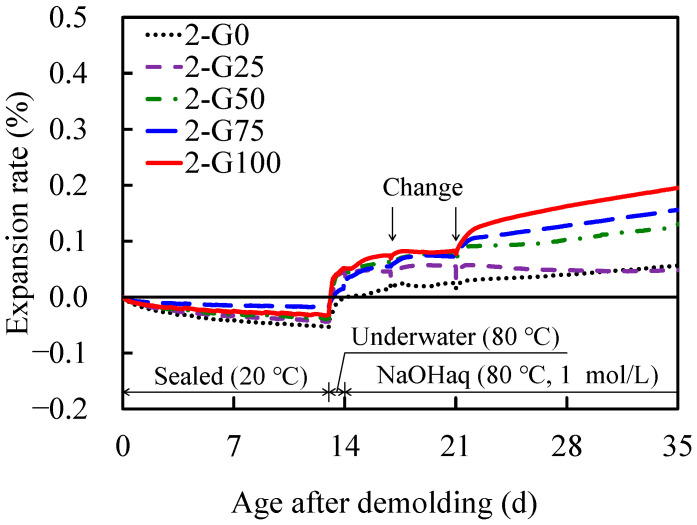
Diurnal changes in expansion rate as a function of demolding age (Series 2).

**Figure 7 materials-17-04902-f007:**
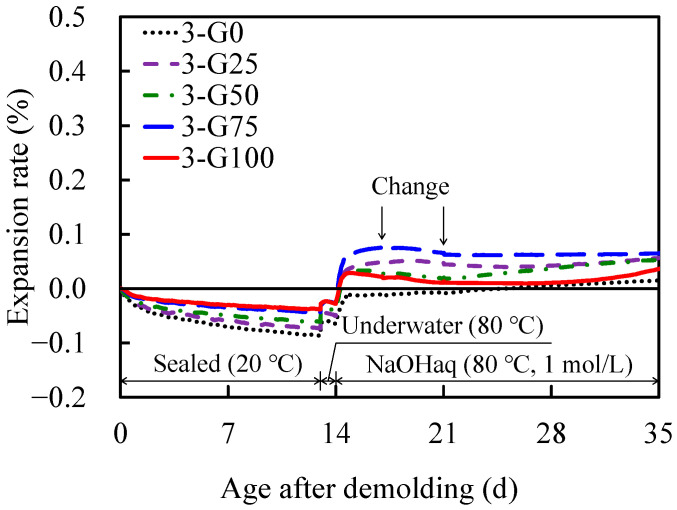
Diurnal changes in expansion rate as a function of demolding age (Series 3).

**Figure 8 materials-17-04902-f008:**
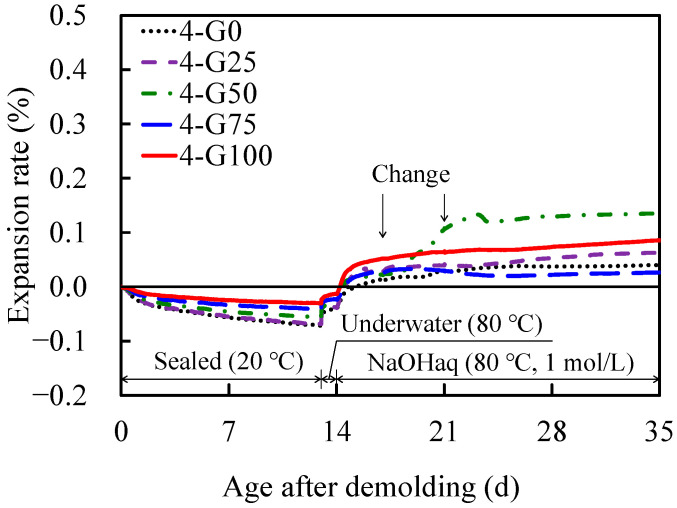
Diurnal changes in expansion rate as a function of demolding age (Series 4).

**Figure 9 materials-17-04902-f009:**
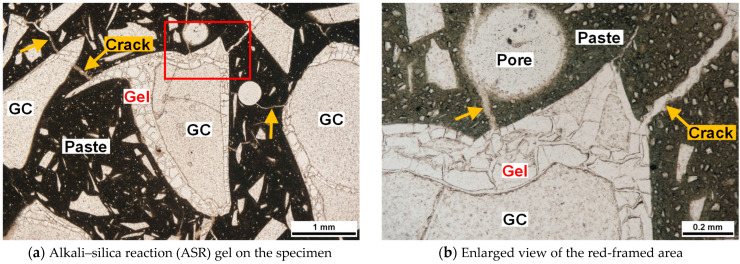
Polarized light microscopy images (single Nicole) of the specimen 1-G100.

**Figure 10 materials-17-04902-f010:**
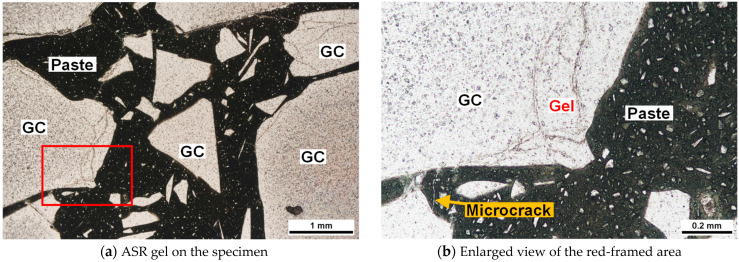
Polarized light microscopy images (single Nicole) of the specimen 3-G100.

**Figure 11 materials-17-04902-f011:**
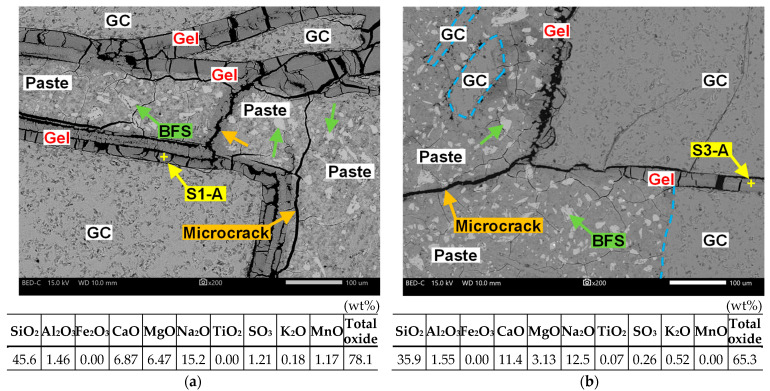
Backscattered electron images of the cut surfaces of the specimens and the chemical compositions of the ASR gels. (**a**) Image of the 1-G100 specimen and chemical compositions on the S1-A gel; (**b**) Image of the 3-G100 specimen and chemical compositions on the S3-A gel.

**Figure 12 materials-17-04902-f012:**
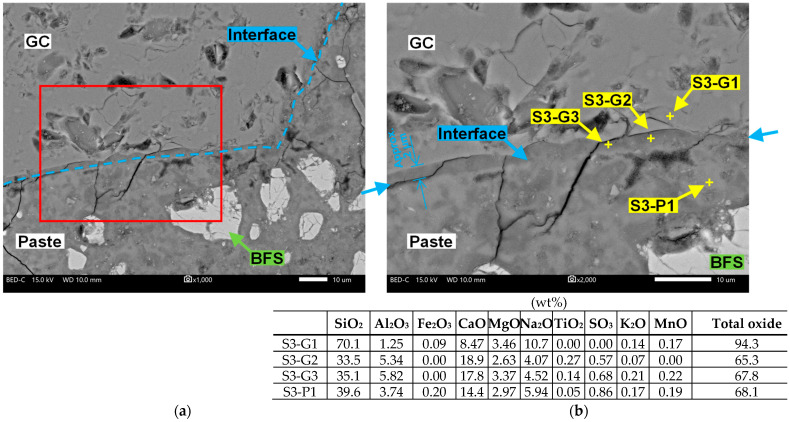
Backscattered electron images of the interface between the GC and the paste in the cut surfaces of the 3-G100 specimen. (**a**) GC and paste interface; (**b**) Enlarged image of the red-framed area and chemical compositions on the GC and the paste.

**Table 1 materials-17-04902-t001:** Raw material specifications.

Material	Abbreviation	Details
Water	W	Tap water (Density: 1.00 g/cm^3^)
Alkali source	SO	Sodium orthosilicate (Na_4_O_4_Si; Density: 2.39 g/cm^3^; Water content: 1.7%)
Precursors	BFS	Blast-furnace slag fine powder 4000 (Density: 2.91 g/cm^3^; Specific surface area: 4160 cm^2^/g; without gypsum addition)
	FA	Fly ash (Density: 2.36 g/cm^3^; Specific surface area: 5650 cm^2^/g)
Fine aggregates	S	Crushed sand (Place of origin: Takagi, Satsumasendai City, Kagoshima Prefecture, Japan; Density: 2.85 g/cm^3^; Fineness modulus: 2.52)
	GC	Photovoltaic panel glass cullet (Density: 2.46 g/cm^3^; Fineness modulus: 4.37)

**Table 2 materials-17-04902-t002:** Chemical compositions of the raw materials used in this study (wt%).

Component	SO	BFS	FA	GC
SiO_2_	29.7	30.5	52.3	70.3
Al_2_O_3_	0.07	13.6	32.3	1.38
Fe_2_O_3_	—	0.25	7.37	—
CaO	0.29	45.9	2.53	11.6
MgO	—	6.37	1.51	3.33
Na_2_O	69.8	—	—	13.0
TiO_2_	—	0.49	1.40	—
SO_3_	0.07	2.01	0.76	0.21
K_2_O	0.07	0.40	1.00	0.03
MnO	—	0.26	—	—
P_2_O_5_	—	—	0.67	—
SrO	—	0.11	0.11	—
BaO	—	0.06	—	—
ZrO_2_	—	0.04	—	—
Y_2_O_3_	—	0.01	—	—
Sb_2_O_3_	—	—	—	0.22

**Table 3 materials-17-04902-t003:** Mix designs for the AAM mortar.

Series	Abbreviations	BFS Ratio in Precursors	GC Ratio inFine Aggregates	W	SO	BFS	FA	S	GC
		(vol%)		(kg/m^3^)					
1 (AAS)	1-G0	100	0	274	63	582	0	1425	0
	1-G25		25					1069	308
	1-G50		50					713	615
	1-G75		75					356	923
	1-G100		100					0	1230
2 (AAS)	2-G0	100	0	273	45	606		1425	0
	2-G25		25					1069	308
	2-G50		50					713	615
	2-G75		75					356	923
	2-G100		100					0	1230
3 (AAS) GC after autoclave process	3-G0	100	0					1425	0
	3-G25		25					1069	308
	3-G50		50					713	615
	3-G75		75					356	923
	3-G100		100					0	1230
4 (AAFS)	4-G0	50	0			303	246	1425	0
	4-G25		25					1069	308
	4-G50		50					713	615
	4-G75		75					356	923
	4-G100		100					0	1230

**Table 4 materials-17-04902-t004:** Inductively coupled plasma analysis results for dissolved constituents.

Chemical Element	Distilled Water	Autoclave Elution Water
	(mg/L)	
Si	<0.1	1300
Na	<0.1	1600
Ca	<0.1	4.8

## Data Availability

All data are contained within the article.

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
