# Peer review of "Expansion Control of Alkali-Activated Materials Using Waste Glass Cullet from Photovoltaic Panels as Fine Aggregates"

_materials, 2024, doi:10.3390/ma17194902_

Round 1

Reviewer 1 Report

Comments and Suggestions for Authors

In this study, AAMs with glass cullet as fine aggregate is investigated. Generally speaking, this is a very interesting paper especially for the glass upon leaching test. The experiments are well designed and the conclusions are supported by the results. However, there are some problems which the authors should look into.

1. In the third paragraph, it is better to clarify geopolymer is low-Ca AAM, which is accepted by recent publications. And I do not know why two terms AAM and GP are used for one thing in this paper.

2. In the second paragraph, the authors should include the most recent studies. There are feasible ways for large volume utilization of glass aggregate in cement concrete such as using calcium nitrate as an additive to suppress Si dissolution rate:. (e.g., Calcium nitrate effectively mitigates alkali-silica reaction by surface passivation of reactive aggregates). 

3. I personally suggest revising Table 3 by reducing the number of terms for better readability. For example water (W), P, A, S etc. are not really necessary. Most readers wouldn't read the paper word by word. Once people get lost, they wouldn't read it anymore.

4. The strength test method should include the loading rate.

5. Previous paper shows ASR expansion first increases with alkali dosage, then decrease. A very high dosage of alkali may contribute to the dissolution of aggregate at early ages, and those dissolved silicon participates in the geopolymerization instead of ASR. A very high dosage of alkali may also thermodynamically hinder ASR. See toward waste glass upcycling: preparation and characterization of high volume waste glass geopolymer composites.

4. The authors used BSE (backscattered imaging). SEM implies it is SE mode not BSE. It is better to use BSE, and BSE-EDS.

Author Response

Dear Reviewer,

We sincerely thank you for taking the time to review our manuscript and for providing valuable comments and suggestions.

We have addressed your comments and revised the manuscript accordingly. Itemized responses to all the listed comments follow. The changes in the manuscript are highlighted in red.

  1. In the third paragraph, it is better to clarify geopolymer is low-Ca AAM, which is accepted by recent publications. And I do not know why two terms AAM and GP are used for one thing in this paper.

It has been indicated in the revised document that the geopolymer is a low-Ca AAM (see lines 72–74).

Additionally, the terms GP and AAM have been unified to AAM.

The symbol list in Table 1 has been added for ease of reading.

  1. In the second paragraph, the authors should include the most recent studies. There are feasible ways for large volume utilization of glass aggregate in cement concrete such as using calcium nitrate as an additive to suppress Si dissolution rate:. (e.g., Calcium nitrate effectively mitigates alkali-silica reaction by surface passivation of reactive aggregates).

It has been indicated in the revised document that carbonation curing and the addition of calcium nitrate are effective in suppressing ASR (see lines 55–58).

  1. I personally suggest revising Table 3 by reducing the number of terms for better readability. For example water (W), P, A, S etc. are not really necessary. Most readers wouldn't read the paper word by word. Once people get lost, they wouldn't read it anymore.

The symbols AW, P, and A used in Table 3 (original manuscript version) (changed to Table 4 in the revised document) have been deleted.

  1. The strength test method should include the loading rate.

It has been noted in the revised manuscript that the loading rate during the compression test is 36 MPa/min (see lines 191–192).

  1. Previous paper shows ASR expansion first increases with alkali dosage, then decrease. A very high dosage of alkali may contribute to the dissolution of aggregate at early ages, and those dissolved silicon participates in the geopolymerization instead of ASR. A very high dosage of alkali may also thermodynamically hinder ASR. See toward waste glass upcycling: preparation and characterization of high volume waste glass geopolymer composites.

Discussion comments referring to the article “Xiao, R.; Dai, X.; Zhong, J. Ma, Y.; Jiang, X.; He, J.; Wang Y.; Huang, B. Toward waste glass upcycling: Preparation and characterization of high-volume waste glass geopolymer composites, SM&T 2024, 40, e00890. https://doi.org/10.1016/j.susmat.2024.e00890” have been added in lines 370–372, 413–416, and 447–449 of the revised document.

  1. The authors used BSE (backscattered imaging). SEM implies it is SE mode not BSE. It is better to use BSE, and BSE-EDS.

As pointed out, these images were observed by using BSE. Therefore, the titles of Figures 11 and 12 have been revised from “Scanning electron microscopy images” to “Backscattered electron images.”

Reviewer 2 Report

Comments and Suggestions for Authors

Dear Authors,

The topic taken up by the authors is important, due to the increasing amount of waste photovoltaic panels. The authors, in addition to glass cullet from photovoltaic panels, also use other waste - fly ash, blast-furnace slag. Developing technologies enabling waste management is desirable.

The text of the manuscript still requires refinement. Mainly: please conduct a scientific discussion of the results - please refer to the results of studies published by other researchers.

Detailed remarks:

Chapter 1. Introduction

Line 48

Quote: „Additionally, it will take approximately 19 years to treat all the PV waste generated in Japan by 2020.”

Comment: Please remember, it is currently 2024.

Chapter 2. Experimental overview  2.1. Materials and specimen preparation

Line 158

Quote: „Water was then added,….”

Comment: How much water was added?

Chapter 2. Experimental overview  2.2. Experimental methods

According to the authors' declaration, the AAM produced was intended for use in precast products, such as road curbs. This means that they will function in the environment. They will be exposed to the interference of rainwater (often with acidic pH), soil water (soil solutions with acidic pH). Therefore, leachability tests should be performed to determine the concentration of heavy metals in the eluates (i.e. the potential load of heavy metals that may end up in the environment). Let me remind you that glass cullet from photovoltaic panels, fly ash, blast-furnace slag were used. Each of these wastes contains heavy metals.

Chapter 3. Results and discussions

Please conduct a scientific discussion of the results - please refer to research results published by other researchers.

3.1. Mortar-flow test

Fig. 3.

Although the differences in flow values ​​are small for GC substitution rate, a trend line is clearly visible for series 4. For values ​​0-50 GC/A there was an increase (0<25<50), and for values ​​50-100 there was a decrease (50>75>100). It should be noted that the value GC/A 100 is lower than 0. Please explain this scientifically.

 Best regards,

Reviewer

Author Response

Dear Reviewer,

We sincerely thank you for taking the time to review our manuscript and for providing valuable comments and suggestions.

We have addressed your comments and revised the manuscript accordingly. Itemized responses to all the listed comments follow. The changes in the manuscript are highlighted in red.

Chapter 1. Introduction

Line 48

Quote: „Additionally, it will take approximately 19 years to treat all the PV waste generated in Japan by 2020.”

Comment: Please remember, it is currently 2024.

We apologize for the mistake. The following comment was corrected in the revised manuscript (lines 49–50): “Additionally, the expected required time for the processing of all PV waste using current technology and processing capacity is of the order of decades.”

Chapter 2. Experimental overview  2.1. Materials and specimen preparation

Line 158 (now changed to line 165)

Quote: „Water was then added,….”

Comment: How much water was added?

The revised phrase indicates that all the water was added.

Chapter 2. Experimental overview  2.2. Experimental methods

According to the authors' declaration, the AAM produced was intended for use in precast products, such as road curbs. This means that they will function in the environment. They will be exposed to the interference of rainwater (often with acidic pH), soil water (soil solutions with acidic pH). Therefore, leachability tests should be performed to determine the concentration of heavy metals in the eluates (i.e. the potential load of heavy metals that may end up in the environment). Let me remind you that glass cullet from photovoltaic panels, fly ash, blast-furnace slag were used. Each of these wastes contains heavy metals.

As a result of the component analysis in Table 3, it was found that the glass cullet, fly ash, and blast furnace slag did not contain heavy metals. This comment has been added in lines 143–144 of the revised manuscript.

Additionally, as pointed out, these materials may contain heavy metals. Therefore, an explanatory comment has been added in lines 452–455 of the revised document indicating that heavy metal elution tests are necessary in future studies.

Chapter 3. Results and discussions

Please conduct a scientific discussion of the results - please refer to research results published by other researchers.

Discussion comments referring to the results of other studies have been added in lines 249–252, 282–283, 370–373, 408–416, and 447–449 of the revised document.

3.1. Mortar-flow test

Fig. 3.

Although the differences in flow values are small for GC substitution rate, a trend line is clearly visible for series 4. For values 0-50 GC/A there was an increase (0<25<50), and for values 50-100 there was a decrease (50>75>100). It should be noted that the value GC/A 100 is lower than 0. Please explain this scientifically.

The following comment was added in the revised manuscript (lines 256–267):

“In Series 4, the flow value increased in the fine aggregate for GC ratios up to 50 % but decreased thereafter. As the flow value is affected considerably by the size and density of the fine aggregate, the fluidity is believed to increase for GC ratio values up to 50 % owing to its replacement with GC (which has a density of 2.46 g/cm3 which is lower than the sand’s density of 2.85 g/cm3). Moreover, as the replacement amount increased, the fluidity decreased owing to the GC (which comprises larger particle sizes than sand), occupying the mortar, thus making it difficult for the GC to move. The reason that this phenomenon did not occur in Series 1, 2, and 3 was presumably because the density of BFS (equal to 2.91 g/cm3)—which is the base of the mortar—is greater than that of FA (density: 2.36 g/cm3); therefore, it is less affected by the aggregate compared with FA. This phenomenon is planned to be clarified in future experiments.”

Reviewer 3 Report

Comments and Suggestions for Authors

The article entitled Expansion Control of Alkali-Activated Materials Using Waste Glass Cullet from Photovoltaic Panels as Fine Aggregates addresses the problem of partial substitution of fine aggregate in alkali-activated material mortars. I consider this topic to be current and important from a scientific and application point of view.

The authors focused on determining such features as: mortar-flow test, compressive strength test, accelerated expansion test. In order to better understand the reactions and dependencies, they took SEM images of the obtained mortar samples. The results of these analyses are discussed in detail, although perhaps a little too little compared to the results of research by other authors. At the same time, the literature review is detailed (47 items) and up-to-date (28 out of 47 items come from the last 5 years). No self-citations were noted. The authors indicated the need to conduct research in this area, but in the final part of the Introduction section, the novelty aspect of this particular work is described a little too generally.

The manuscript has a clear, legible structure, the figures and tables are easy to interpret and of good quality. The authors described the planned experiment in detail, it is possible to reproduce it by other authors. The conclusions are consistent with the obtained results, presented in a legible and concise manner.

Detailed comments:

1. In my opinion, this work lacks any statistical analysis. The authors wrote that they performed the compressive strength test on 3 samples for each case. Perhaps it would be worth including error bars (standard deviation) on the graphs, which would allow for an assessment of repeatability, whether the results are reliable.

2. The text lacks a reference to item [43].

Author Response

Dear Reviewer,

We sincerely thank you for taking the time to review our manuscript and for providing valuable comments and suggestions.

We have addressed your comments and revised the manuscript accordingly. Itemized responses to all the listed comments follow. The changes in the manuscript are highlighted in red.

  1. In my opinion, this work lacks any statistical analysis. The authors wrote that they performed the compressive strength test on 3 samples for each case. Perhaps it would be worth including error bars (standard deviation) on the graphs, which would allow for an assessment of repeatability, whether the results are reliable.

Error bars have been added to Figure 4. Additionally, on lines 267–268 of the revised manuscript, a comment has been added indicating that the plotted values are the means and that the error bars denote the standard deviation (above and below the mean).

  1. The text lacks a reference to item [43].

There was an error in the reference number on line 273. It erroneously cited [44] instead of [43]. We apologize for the mistake. Given all the manuscript’s revisions, this citation has been changed to [49].

Reviewer 4 Report

Comments and Suggestions for Authors

Some issues need to be solved before its consideration, as a publishable document in the journal, described as follows:

+ How do the authors select in % the independent variables as the source of the mixture design? By convenience?

+ Standards need to be cited in the document. Then how many tests were performed by mix?

+ A symbology list can be useful.

+ From Figure 3, what are the units for Flow Value?

+ Provide samples as they were made and after tests.

+ It's important to compare and discuss the results obtained with the consulted documents. For example, the information in lines 410 to 412 is so vague. Check similar paragraphs throughout the document.

+ The conclusions are not aligned with the objectives. Therefore, it is necessary to clearly describe the purpose of the study so that the result points are the same as the conclusions.

+ Conclusions in accordance with the objectives set, however, it is desirable to expand the text in them.

+ It is recommended to describe the results obtained with add a scientific interpretation of the results.

Comments on the Quality of English Language

some issues required revisions

Author Response

Dear Reviewer,

We sincerely thank you for taking the time to review our manuscript and for providing valuable comments and suggestions.

We have addressed your comments and revised the manuscript accordingly. Itemized responses to all the listed comments follow. The changes in the manuscript are highlighted in red.

+ How do the authors select in % the independent variables as the source of the mixture design? By convenience?

The target mixing ratio of GC to fine aggregate was 100 % because we wanted to use all of the GC. However, as it is considered that at a target mixing ratio of 100 %, the compressive strength will be insufficient, the flow value will not meet the target (≥180 mm), and the impact of ASR expansion in particular will be large, we conveniently selected values from 0 % to 100 % in 25 % increments.

In addition, the proportions of BFS and FA in the precursor were set to 0 % and 50 %, respectively. BFS develops strength faster than FA and does not require a supply of heat. Thus, it can be manufactured in small precast factories that do not have steam curing equipment; therefore it will be based on BFS. However, because BFS contains more Ca than FA and it is considered that the impact of ASR expansion will be large, 50 % of the precursor was replaced with FA for convenience.

+ Standards need to be cited in the document. Then how many tests were performed by mix?

The numbers of trials for each experiment have been added in lines 182–185 and 190–191 of the revised manuscript.

+ A symbology list can be useful.

A list of symbols has been added in Table 1 of the revised manuscript.

+ From Figure 3, what are the units for Flow Value?

The unit of the length of the spread of the mortar after the removal of the cylinder is in [mm]; relevant changes have been implemented in Figures 3.

+ Provide samples as they were made and after tests.

We assume your comment refers to the inclusion of polarized light microscopy and SEM images of the specimen before and after ASR expansion.

We imaged the specimens with an optical microscope before the accelerated expansion test. Figure 1 (shown in the word file) shows the images of the cut surfaces of the specimens, but the presence or absence of ASR expansion can be determined even with an optical microscope. However, as it was not possible to observe the aggregate interface, we imaged the specimens with polarizing and scanning electron microscopy. However, as the use of these imaging modalities is very expensive, we decided to select only the two samples 1-100 and 3-100; relevant results obtained from these samples are included in the revised manuscript.

We agree that is better to publish sample images before the occurrence of ASR. Accordingly, we will publish these in future publications.

+ It's important to compare and discuss the results obtained with the consulted documents. For example, the information in lines 410 to 412 is so vague. Check similar paragraphs throughout the document.

Discussion comments referring to the results of other studies have been added in lines 249–252, 282–283, 370–373, 408–416, and 447–449 of the revised document.

The information on lines 410-412 has been corrected and moved to lines 425–427.

+ The conclusions are not aligned with the objectives. Therefore, it is necessary to clearly describe the purpose of the study so that the result points are the same as the conclusions.

In response, the purpose of the research is clearly stated in lines 111–121.

+ Conclusions in accordance with the objectives set, however, it is desirable to expand the text in them.

The conclusions in lines 463–506 have been expanded based on the stated objectives.

+ It is recommended to describe the results obtained with add a scientific interpretation of the results.

The results were examined in-depth and relevant scientific interpretations were provided.

Round 2

Reviewer 1 Report

Comments and Suggestions for Authors

This paper has been revised and should be ready for publication.

Author Response

Dear Reviewer,

We sincerely thank you for taking the time to review our manuscript a second time.

This paper has been revised and should be ready for publication.

The manuscript has been revised and benefitted from your thoughtful suggestions and insights.

Reviewer 2 Report

Comments and Suggestions for Authors

Dear Authors,

I would like to thank the Authors for the available and accessible manuscript. The quality of the manuscript is definitely there.

I have some minor comments on the inserted text.

Lines 143-144

Added: "Moreover, these raw materials did not contain heavy metals".

Comment: This statement cannot be agreed with. Waste - glass cullet from photovoltaic panels, fly ash, blast furnace slag always contain some metal load. If the authors did not perform determinations of these metals in the waste (which will be used as raw materials), it should be written that it will be recorded in the course of the research.

 Lines 182-183

 Added: "It is common practice to perform the research twice and adopt the average value of the results as the flow value.

Comment: This is a negative term. Three measurements should always be used to eliminate gross errors.

Best regards,

Reviewer

Author Response

Dear Reviewer,

We sincerely thank you for taking the time to review our manuscript and for providing valuable comments and suggestions for a second time. The quality of the manuscript has improved because of your valuable insights. We have addressed your comments and have revised the manuscript accordingly. Itemized responses to all of the listed comments are as follows, and changes in the manuscript have been highlighted in blue.

Lines 143-144

Added: "Moreover, these raw materials did not contain heavy metals".

Comment: This statement cannot be agreed with. Waste - glass cullet from photovoltaic panels, fly ash, blast furnace slag always contain some metal load. If the authors did not perform determinations of these metals in the waste (which will be used as raw materials), it should be written that it will be recorded in the course of the research.

Thank you for pointing this out to us. X-ray fluorescence diffraction measurements were performed again on the glass cullet from photovoltaic panels. The solar glass cullet contained 0.22 wt% antimony, which has been added to Table 3. Therefore, the statements in Lines 142 and 476–477, alluding to not containing any heavy metals, were deleted.

 Lines 182-183

 Added: "It is common practice to perform the research twice and adopt the average value of the results as the flow value.

Comment: This is a negative term. Three measurements should always be used to eliminate gross errors.

My sincerest apologies for the misunderstanding on my part, which led to my poor explanation. The flow test was performed thrice. What we wanted to convey was that in one test, the spread of the mortar was measured from two directions and the average value was used. Therefore, we have revised Lines 176–181 as follows:

“To measure the mortar flowability, a flow test was conducted based on Japanese Industrial Standards (JIS) R 5201, where the mortar was packed into a flow cone (upper inner diameter: 70 mm, lower inner diameter: 100 mm, and height: 60 mm) immediately after mixing. After removing the cylinder, the length of the mortar that spread with 15 falling movements was measured in two directions and the average was calculated. This procedure was performed thrice, and the average of these values ​​was used as the flow.”

Reviewer 4 Report

Comments and Suggestions for Authors

Thank you to the authors for addressing my queries. The revisions have significantly improved the manuscript. However, before publication, a few additional issues need to be addressed as outlined below:

The list of symbols or abbreviations should be placed at the end of the manuscript (please refer to the author guidelines).

The results should be compared and discussed in relation to similar studies.

No further comments, good job

Author Response

Dear Reviewer,

We sincerely thank you for taking the time to review our manuscript a second time and for providing valuable comments and suggestions. We have addressed your comments and revised the manuscript accordingly. Itemized responses to all the listed comments follow, and the changes in the manuscript are highlighted in blue.

The list of symbols or abbreviations should be placed at the end of the manuscript (please refer to the author guidelines).

The following text in Line 83 has been moved to Line 536:

“The abbreviations used in this article and their meanings are listed in Table A.”

Table A (changed from Table 1) has been moved to the end of the manuscript (Line 537); therefore, the table number has been shifted accordingly.

The results should be compared and discussed in relation to similar studies.

A comparison and discussion of the accelerated expansion test results was made with a similar study: “Wang, W.; Maruyama, I.; Noguchi, T. Impact of exposure conditions on alkali-silica reaction in alkali-activated material systems. Cem. Concr. Compos. 2024, 153, 105695. https://doi.org/10.1016/j.cemconcomp.2024.105695.” This has been added to Lines 359–374 and 383–385 of the revised manuscript.
